# Infectome analysis of bat kidneys from Yunnan province, China, reveals novel henipaviruses related to Hendra and Nipah viruses and prevalent bacterial and eukaryotic microbes

Guopeng Kuang[1⊛], Tian Yang[1,2⊛] Weihong Yang[1], Jing Wang[3,4,5], Hong Pan[1], Yuanfei Pan[6], Qin-yu Gou[3,4,5], Wei-chen Wu[3,4,5], Juan Wang[1], Lifeng Yang[1], Xi Han[1], Yao-qing Chen[7], John-Sebastian Eden[8], Edward C. Holmes[8,9], Mang Shi[3,4,5]*, Yun Feng[1,2,10]*

1 Yunnan Provincial Key Laboratory for Zoonosis Control and Prevention, Yunnan Institute of Endemic Disease Control and Prevention, Kunming, China, 2 School of Public Health, Dali University, Dali, China, 3 National Key Laboratory of Intelligent Tracking and Forecasting for Infectious Diseases, Shenzhen Campus of Sun Yat-sen University, Sun Yat-sen University, Shenzhen, China, 4 State Key Laboratory for Biocontrol, Shenzhen Campus of Sun Yat-sen University, Sun Yat-sen University, Shenzhen, China, 5 Shenzhen Key Laboratory for Systems Medicine in Inflammatory Diseases, School of Medicine, Shenzhen Campus of Sun Yat-sen University, Sun Yat-sen University, Shenzhen, China, 6 Ministry of Education Key Laboratory of Biodiversity Science and Ecological Engineering, School of Life Sciences, Fudan University, Shanghai, China, 7 School of Public Health (Shenzhen), Shenzhen Campus of Sun Yat-sen University, Sun Yat-sen University, Shenzhen, China, 8 School of Medical Sciences, The University of Sydney, Sydney, New South Wales, Australia, 9 Laboratory of Data Discovery for Health Limited, Hong Kong SAR, China 10 State Key Laboratory of Remote Sensing Science, Center for Global Change and Public Health, Faculty of Geographical Science, Beijing Normal University, Beijing, China

⊛ These authors contributed equally to this work.
* ynfy428@163.com (YF); shim23@mail.sysu.edu.cn (MS)

## Abstract

Bats are natural reservoirs for a wide range of microorganisms, including many notable zoonotic pathogens. However, the composition of the infectome (i.e., the collection of viral, bacterial and eukaryotic microorganisms) within bat kidneys remains poorly understood. To address this gap, we performed meta-transcriptomic sequencing on kidney tissues from 142 bats, spanning ten species sampled at five locations in Yunnan province, China. This analysis identified 22 viral species, including 20 novel viruses, two of which represented newly discovered henipaviruses closely related to the highly pathogenic Hendra and Nipah viruses. These henipaviruses were found in the kidneys of bats inhabiting an orchard near villages, raising concerns about potential fruit contamination via bat urine and transmission risks to livestock or humans. Additionally, we identified a novel protozoan parasite, tentatively named *Klossiella yunnanensis*, along with two highly abundant bacterial species, one of which is a newly discovered species—*Flavobacterium yunnanensis*. These findings broaden our understanding of the bat kidney infectome, underscore critical zoonotic threats, and highlight the need for comprehensive, full-spectrum microbial

**Data availability statement:** The meta-transcriptomic sequencing reads generated in this study have been deposited in the NCBI Sequence Read Archive (SRA) database under BioProject accession PRJNA1184956. The whole genome sequences of viruses generated in this study are available at NCBI/GenBank under accession numbers PQ621837 to PQ621860, PQ815815, and PQ824231.

**Funding:** This study was funded by grants from the National Key R&D Program of China (2024YFC2607501 & 2024YFC2607502 to M.S.), Yunnan Revitalization Talent Support Program Top Physician Project (XDYC-MY-2022-0074 to Y.F.), the National Natural Science Foundation of China (82341118 to M.S.), Natural Science Foundation of Guangdong Province of China (2022A1515011854 to M.S.), Shenzhen Science and Technology Program (KQTD20200820145822023 to M.S.), Major Project of Guangzhou National Laboratory (GZNL2023A01001 to M.S.), Guangdong Province "Pearl River Talent Plan" Innovation, Entrepreneurship Team Project (2019ZT08Y464 to M.S.), and the Fund of Shenzhen Key Laboratory (ZDSYS20220606100803007 to M.S.), National Health & Medical Research Council (NHMRC) Investigator grant (GNT2017197 to E.C.H.) and AIR@InnoHK administered by the Innovation and Technology Commission, Hong Kong Special Administrative Region, China (to E.C.H.). The funders had no role in study design, data collection and analysis, decision to publish, or preparation of the manuscript.

**Competing interests:** The authors have declared that no competing interests exist.

analyses of previously understudied organs to better assess spillover risks from bat populations.

## Author summary

Although extensive investigations have been conducted on the bat virome, most studies have focused on fecal samples, leaving other tissues, such as the kidney, largely unexplored. However, the kidney can harbor important zoonotic pathogens, including the highly pathogenic Hendra and Nipah viruses, and genomic evidence of henipaviruses in bats from China has remained undocumented. In this study, we report the first detection of two novel henipavirus genomes from bat kidneys in China, one of which is the closest known relative of pathogenic henipaviruses identified to date. Beyond virome analysis, our study also examined highly prevalent bacteria and eukaryotic microbes, identifying those potentially relevant to bat infections. Overall, these findings provide valuable insights into the infectome of the bat kidney, highlighting the need for broader microbial surveillance beyond the gastrointestinal tract.

## Introduction

Bats (order *Chiroptera*) are one of the most diverse and abundant groups of mammals, comprising nearly 1,500 species with a near global distribution [1]. Bats are also well-known natural reservoirs for a wide variety of microbial pathogens, a characteristic often attributed to their unique immune systems which maintain a delicate balance between host defenses and immune tolerance to viral infections [2–4]. Importantly, bats have been implicated in a number of major emerging disease outbreaks, including Hendra [5], Nipah [6], Marburg and Ebola [7] virus disease, severe acute respiratory syndrome (SARS) [8], Middle East respiratory syndrome (MERS) [8], and coronavirus disease 2019 (COVID-19) [9]. Indeed, comparative studies indicate that bats harbor a greater diversity of viruses than many other mammalian groups, underscoring their significance for zoonotic disease surveillance [10].

Metagenomic approaches have greatly advanced the characterization of bat viromes, deepening our understanding of the diversity of bat-borne pathogens and their potential role in disease emergence and transmission [11–14]. As of October 2024, viral sequences from at least 31 families have been identified in 340 bat species across 111 countries [15]. Bat-borne viruses are transmitted to humans either through direct contact with bats or via so-called "intermediate" hosts, often linked to the ingestion of food or water contaminated with bat saliva, feces, or urine [16]. Although most research has concentrated on the bat gut virome, viruses residing in other tissues, including the kidneys where they may be excreted via urine, also present potential transmission risks. Indeed, zoonotic viruses have been detected in bat kidneys and urine, including henipaviruses [17–20], pararubulaviruses [20–25], and

betacoronaviruses [25]. As these kidney-associated pathogens can be excreted through urine they are at heightening risk of human exposure.

Beyond viruses, bats harbor a diverse array of bacteria, fungi, and protozoan parasites that infect bats or even humans [26,27]. A notable example is the psychrophilic fungus *Pseudogymnoascus destructans*, which has caused a devastating disease in bats and led to the deaths of millions of animals across eastern North America [27]. Although this fungus is not known to pose a direct threat to humans, disturbances during bat hibernation—such as flying during the day and gathering near cave and mine entrances in winter—may increase human-bat encounters. Additionally, zoonotic bacteria and protozoan parasites, such as members of *Leptospira* [28,29] and *Toxoplasma* [30], have been identified in bat kidneys. However, as with viruses, research on the bacteria and eukaryotic pathogens in bat kidneys remain sparse, highlighting a critical gap in our understanding of the diversity of bat pathogens.

Yunnan province, located in southwestern China and bordering a number of Southeast Asian countries, is renowned as a hotspot for bat diversity and bat-borne viral pathogens, including close relatives of Marburg virus [31], SARS-CoV [32,33], and SARS-CoV-2 [9,13,34]. Herein, we utilized a meta-transcriptomics approach to investigate the total infectome—comprising viruses, bacteria, and eukaryotic microbes—in bat kidneys collected from this geographic region. We further identified and characterized potential human pathogens of notable zoonotic risk and explored interactions between viruses and their protozoan parasite hosts, offering valuable insights into the complexity of the bat kidney infectome.

## Results

### Bat species identification

Between 2017 and 2021, kidney tissues were sampled from 142 individual bats across five cities/counties in Yunnan province, China (Fig 1A and S1 Table). Species identification was initially performed by recovering partial cytochrome c oxidase I (*cox1*) gene sequences using targeted PCR assay and Sanger sequencing. Phylogenetic analysis of full-length *cox1* sequences, generated from meta-transcriptomic sequencing, confirmed the presence of ten bat species spanning

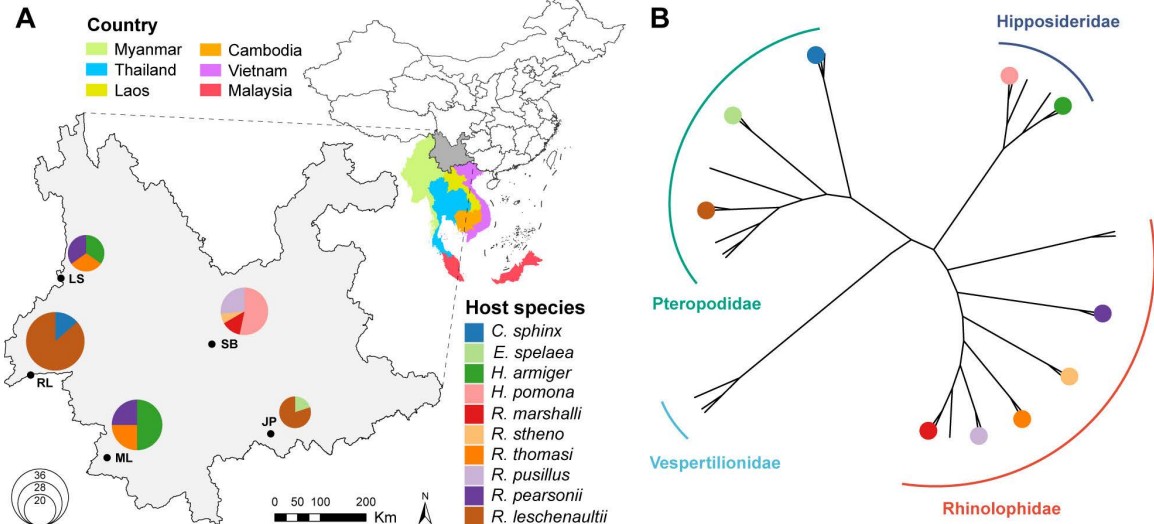

**Fig 1. Bat kidney sampling and species identification.** (A) Map showing the five sampling locations in Yunnan province, China, with nearby countries (Myanmar, Laos, Vietnam, Thailand, Cambodia, and Malaysia) shown for reference. Pie charts indicate the species composition of the bats sampled at each site. The basemap shapefile used in ArcGIS was obtained from the publicly available GADM data set (https://gadm.org/download_country.html). (B) Unrooted phylogenetic tree inferred from full-length *COX1* gene sequences of bat kidney samples analyzed in this study. Colors correspond to different bat species, matching the color scheme used in the pie charts. Branch lengths are scaled to the number of nucleotide substitutions per site.

five genera and three families (Fig 1B and S1 Table). Based on mitochondrial sequences and sampling locations, the samples were pooled into 20 groups for sequencing library constructions, with each group containing 2–8 individuals (S1 Table). Meta-transcriptomic sequencing of total RNA extracted from these pools yielded an average of 56.33 million clean non-rRNA reads, totaling approximately 1.13 billion clean non-rRNA reads.

## Overview of the Yunnan bat kidney infectome

Meta-transcriptomic analysis of the bat kidneys identified a diverse microbial community (Fig 2 and S2 Table). Based on our detection criteria (see Methods), microbes were detected in 18 of the 20 libraries analyzed, comprising 0.06% to 1.28% of total clean non-rRNA reads per library (Fig 2A). Two libraries—one from *Hipposideros armiger* (8 individuals, LS) and another from *Rhinolophus stheno* (2 individuals, SB)—showed no microbial presence. RNA viruses dominated the microbial community, with 20 species from 12 families identified, as well as one DNA virus, one reverse transcribing virus, two bacterial species, and one eukaryotic species (Fig 2B). Among these, the eukaryote genus *Phyllobacterium* was the

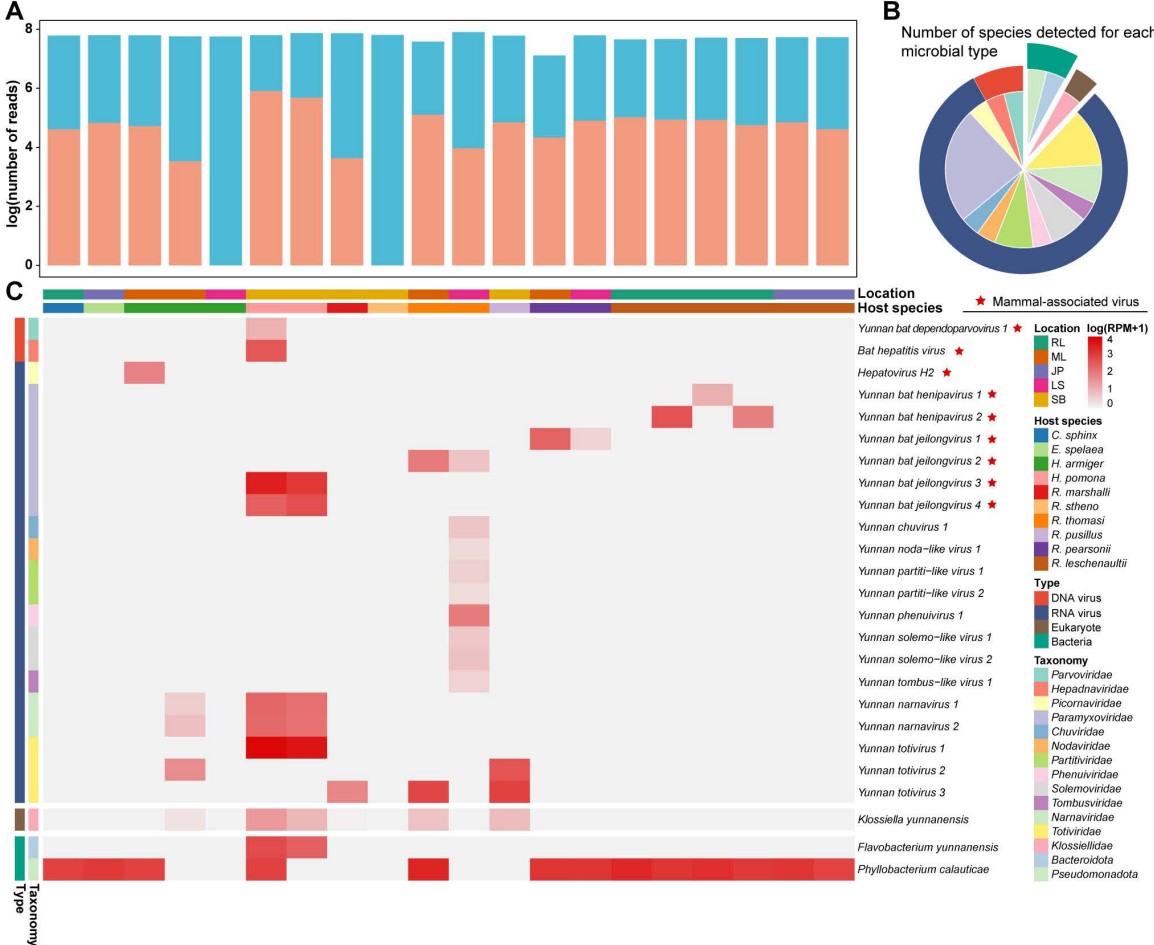

**Fig 2. Overview of the bat kidney infectome.** (A) Numbers of total reads (light blue) and microbial reads (orange) for each library. (B) Number of viral, bacterial, and eukaryotic microbial species detected, with color schemes corresponding to those used in panel C. (C) Heatmap illustrating the distribution and relative abundance of viral, bacterial, and eukaryotic microbes, represented as RNA abundance (RPM: reads per million non-rRNA reads) in each library. Host species and orders are labeled at the top and color-coded according to their respective categories.

most frequently detected, present in 13 (65%) libraries. Notably, one library contains a total of 12 microbial species, all of which were viruses (Fig 2C).

## Virome of bat kidneys

We identified 22 viral species across 12 families in bat kidneys (Figs 2 and 3). These included six RNA viruses from the *Paramyxoviridae*, three from the *Totiviridae*, two each from the *Partitiviridae*, *Solemoviridae*, and *Narnaviridae*, and one each from the *Phenuiviridae, Chuviridae, Nodaviridae, Picornaviridae*, and *Tombusviridae*. Additionally, we discovered one DNA virus from the *Parvoviridae* and one reverse transcribing virus from the *Hepadnaviridae* (Figs 2 and 3). Of these, 20 species (90.91%) spanning 10 families were newly identified per ICTV (International Committee on Virus Taxonomy) species demarcation criteria (S2 Table).

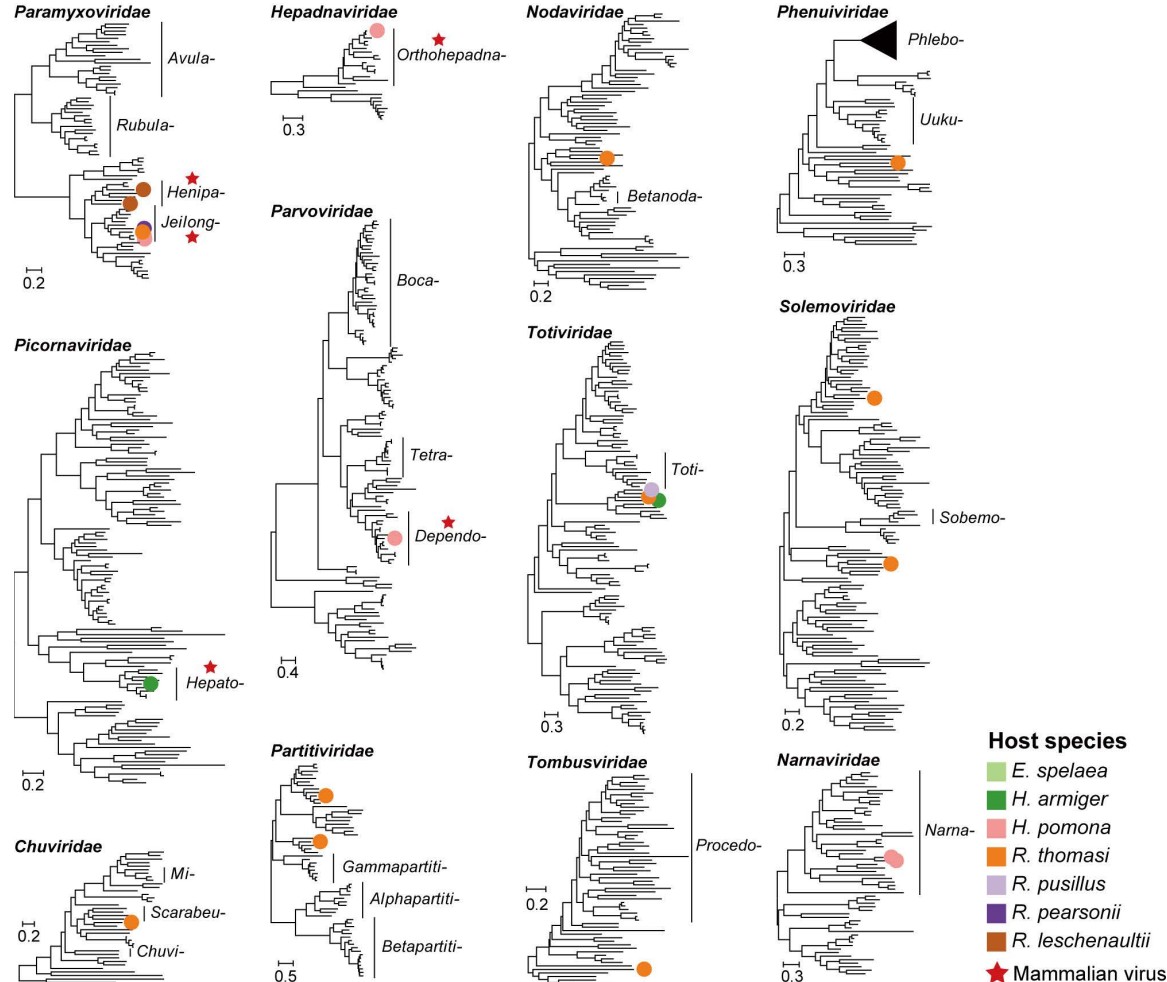

**Fig 3. Phylogenetic diversity of viruses identified in this study.** Phylogenetic trees of viruses from 12 virus families estimated using the maximum likelihood method based on conserved protein sequences (RdRp for RNA viruses, NS1 for *Parvoviridae*, and DNA polymerase for *Hepadnaviridae*). Colored dots on the trees, corresponding to host genera as indicated in the legend, represent viral species identified in this study. Red stars mark members of known mammal-associated viral lineages. All trees are mid-pointed rooted for clarity only with horizontal branch lengths depicting the number of amino acid substitutions per site.

Phylogenetic analyses revealed that nine species (40.91%) were related to known mammal-associated viruses, representing one reverse transcribing virus, one DNA virus and seven RNA viruses (Fig 3). The *Paramyxoviridae* exhibited the highest diversity, with two species from the genus *Henipavirus* and four from the genus *Jeilongvirus* identified. Notably, the two newly identified henipaviruses had a relatively close evolutionary relationship two human pathogens—Hendra virus (HeV, 52.23–56.94% amino acid identity in the L protein) and Nipah virus (NiV, 52.17–57.03% amino acid identity) (Fig 3). In addition, we identified a hepatotropic virus (*Hepadnaviridae,* genus *Orthohepadnavirus*), denoted Bat hepatitis virus variant YNBS16, in bat kidneys. Phylogenetic analysis revealed that this sequence was closely related to an hepadnavirus sequence, ZYPR16 (Clade BtHBV 7), previously identified in bat livers (S1 Fig).

There was also considerable variation in diversity among viral families (Figs 2B, C and 3). While the *Paramyxoviridae* dominated the samples obtained, in many cases the highest abundance members of this family were not associated with the infection of vertebrates, including Yunnan narnavirus 1 and 2 (*Narnaviridae*) and Yunnan totivirus 1–4 (*Totiviridae*). Yunnan totivirus 1 was especially abundant in pools YNBS16 (RPM = 7393.98) and YNBS17 (RPM = 4171.90) (Fig 2C), indicating that its presence was unlikely due to environmental contamination or dietary origin.

## Characterization of newly identified henipaviruses

Among the viruses identified, we focused on those with potential emergence risks based on their phylogenetic relationship to known high-impact human pathogens, specifically Yunnan bat henipavirus 1 and 2. Of the 20 pooled libraries, one (YNBS03) was positive for Yunnan bat henipavirus 1, while two (YNBS02 and YNBS04) contained reads corresponding to Yunnan bat henipavirus 2. These positive pools were all derived from the kidneys of *Rousettus leschenaultii* bats inhabiting an orchard near villages in RL (WD) (Fig 2C).

Using henipavirus genome sequences assembled from these libraries, primers were designed to further examine individual kidneys through qRT-PCR. The results revealed that one kidney from pool YNBS03 (sample WDBS1745), one from YNBS02 (sample WDBS1733), and two from pool YNBS04 (samples WDBS1762 and WDBS1769) tested positive for henipavirus. Further testing of other organs (heart, liver, lung, gut, and brain) from the same individuals (WDBS1733 and WDBS1745) using qRT-PCR and meta-transcriptomic sequencing confirmed the multi-organ presence of henipaviruses within these bats, with the exception of brain tissues (Table 1). Notably, the kidneys exhibited significantly higher viral abundance compared to other organs, suggesting that they are the primary site of henipavirus replication within the host.

The complete genomes of Yunnan bat henipavirus 1 and 2 were successfully assembled from individual kidney samples WDBS1745 and WDBS1733, achieving mean sequencing depths of 27.99X and 1,274.77X, respectively (Fig 4A).

**Table 1. Detection of henipavirus in various organs within individual bats.**

| Library | Organ | Virus query | Length (bp) | Number of reads | qRT-PCR (Ct) | Nested- PCR |
|---------|-------|-------------|-------------|-----------------|--------------|-------------|
| WDBN1745 | brain | Yunnan bat henipavirus 1 | 19755 | 0 | NoCt | – |
| WDBX1745 | heart | Yunnan bat henipavirus 1 | 19755 | 0 | NoCt | – |
| WDBG1745 | liver | Yunnan bat henipavirus 1 | 19755 | 0 | NoCt | – |
| WDBF1745 | lung | Yunnan bat henipavirus 1 | 19755 | 16 | 33.35 | – |
| WDBC1745 | gut | Yunnan bat henipavirus 1 | 19755 | 0 | NoCt | – |
| WDBS1745 | kidney | Yunnan bat henipavirus 1 | 19755 | 3953 | 23.82 | + |
| WDBN1733 | brain | Yunnan bat henipavirus 2 | 17723 | N/A | NoCt | – |
| WDBX1733 | heart | Yunnan bat henipavirus 2 | 17723 | 16 | 31.37 | + |
| WDBG1733 | liver | Yunnan bat henipavirus 2 | 17723 | 15 | 26.04 | + |
| WDBF1733 | lung | Yunnan bat henipavirus 2 | 17723 | 20 | 30.34 | + |
| WDBC1733 | gut | Yunnan bat henipavirus 2 | 17723 | 29 | 30.74 | + |
| WDBS1733 | kidney | Yunnan bat henipavirus 2 | 17723 | 161657 | 16.91 | + |

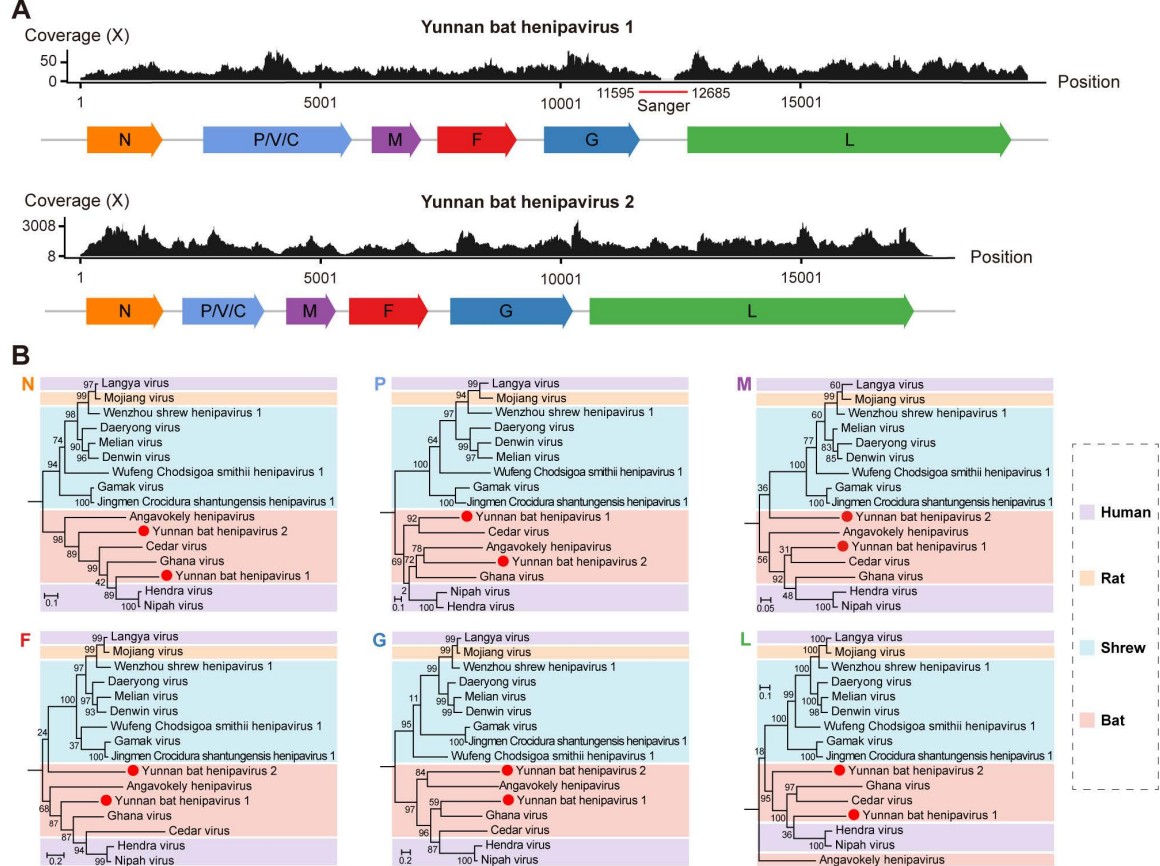

**Fig 4. Characterization of the novel henipavirus species examined in this study.** (A) Genome organization and sequencing coverage of two novel henipavirus species. Coverage across the full-length genome is displayed, with open reading frames (ORFs) depicted as colored arrows below the coverage plots. Regions confirmed by Sanger sequencing for Yunnan bat henipavirus 1 are marked with a red bar beneath the coverage graph. (B) Maximum likelihood phylogenetic trees estimated using amino acid sequences of each gene within the genus *Henipavirus*, rooted with J-virus. Color blocks indicate different species groups, and newly identified viruses are marked with solid red circles. All trees are mid-pointed rooted for clarity only with horizontal branch lengths depicting the number of amino acid substitutions per site.

These two sequences were designated as Yunnan bat henipavirus 1 variant WDBS1745 and Yunnan bat henipavirus 2 variant WDBS1733. The open reading frames (ORFs) and gene arrangements of both viruses were consistent with other members of the genus *Henipavirus*, with each encoding six proteins (Fig 4A).

Phylogenetic analysis of all six genes revealed a clear separation between the predominantly rodent-associated and bat/human-associated clades of the genus *Henipavirus* (Fig 4B). Notably, the newly identified viruses formed distinct lineage, generally grouping with other bat-hosted henipaviruses, including the zoonotic pathogens HeV and NiV, both known for their high mortality rates in humans [35] Yunnan bat henipavirus 1 was most closely related to HeV and NiV in the N (70.33–71.33% amino acid identity) and L proteins (56.94–57.03% amino acid identity), which underscores its potential risk as an emerging pathogen (Fig 4B). However, the phylogenetic positions of Yunnan bat henipavirus 1 and 2 showed marked variability. In particular, Yunnan bat henipavirus 1 was most closely related to HeV and NiV in the N and L proteins trees, but occupied variable positions in the other trees. Although the bootstrap support for these groupings was generally weak, the topological movement of Yunnan bat henipavirus 1 among the henipaviruses likely reflects the action of recombination. Alternatively, the topological inconsistency may result from high sequence divergence, leading to substitution

saturation, increased phylogenetic noise, and hence fewer informative sites for reliable inference. Conversely, the phylogenetic positions of Yunnan bat henipavirus 2 was more consistent across gene trees and also exhibited a greater divergence from other bat henipaviruses.

## Identification and characterization of abundant bacteria in bat kidneys

Our meta-transcriptomic analysis revealed the presence of two relatively abundant bacterial taxa, *Flavobacterium* and *Phyllobacterium*, with conserved marker genes (*rpoB*, *groEL*, *recA*, and *gyrB*) identified via the BLASTx analysis of assembled contigs (S3 Table). Both taxa were represented by assembled contigs with high sequence coverage and depth, enabling the confident reconstruction of marker genes for phylogenetic analysis. Specifically, the representative *rpoB* gene of *Flavobacterium* was assembled from group YNBS16, with a mean coverage of 76.8% and an average depth of 3.30X, while the representative *groEL* gene of *Phyllobacterium* was recovered from group YNBS01, with 91.9% coverage and a mean depth of 5.29X. Phylogenetic analysis revealed that the *Flavobacterium* species forms a distinct branch closely related to *Flavobacterium ammonificans* (94.12% nucleic acid identity in the *ropB* gene) (Fig 5A). This bacterium was tentatively classified as a novel species and named *Flavobacterium yunnanensis*. Similarly, the *Phyllobacterium* species was confirmed as *Phyllobacterium calauticae* based on 97.30% nucleic acid identity and phylogenetic placement of the *groEL* gene (Fig 5B). Further transcriptomic profiling across all 20 pools demonstrated diverse gene expression patterns for these bacteria (Fig 5C), indicating that they are metabolically active within the bat hosts.

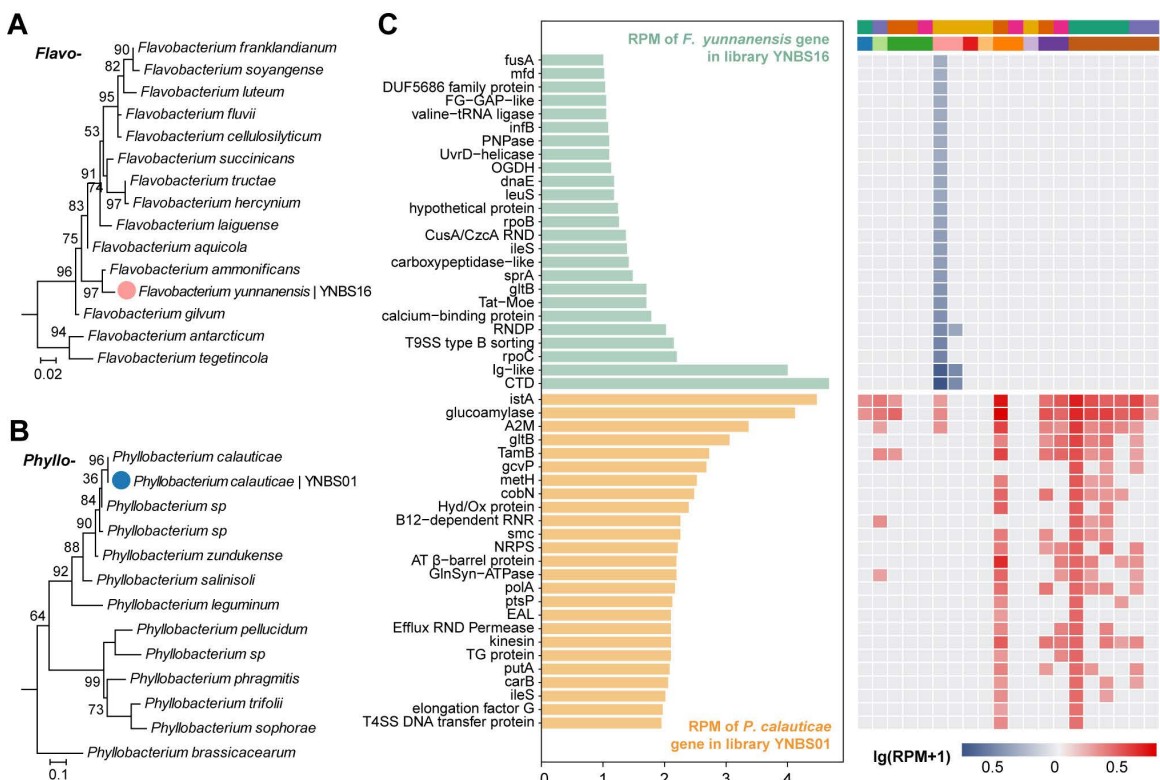

**Fig 5. Gene expression profiles, prevalence and identification of the two bacterial microbes.** (A) Maximum likelihood phylogenetic tree of the genus *Flavobacterium*, constructed using the rpoB gene. (B) Maximum likelihood phylogenetic tree of the genus *Phyllobacterium*, constructed using the groEL gene. (C) Top 25 expressed genes (measured in RPM) for *Flavobacterium yunnanensis* and *Phyllobacterium calauticae* in pools YNBS16 and YNBS01, respectively (left panel), compared with their expression in other positive pools (right panel).

## Eukaryotic microbe identified in bat kidneys

Analysis of the *cox1* and cytochrome b (*cytB*) genes identified a protozoan microbe closely related to the *Klossiella equi* of the family Klossiellidae (phylum Apicomplexa), known to infect the kidney of horses [36]. Phylogenetic and sequence divergence analyses revealed 87.7% nucleotide identity to *K. equi* (MH203050.1) in the *COX1* gene and 91.4% identity in the *cytB* gene. Based on these findings, the newly identified protozoan was proposed as a novel species, tentatively named *Klossiella yunnanensis* (Fig 6A, B). *Klossiella* mitochondrial reads were detected in six pools, exhibiting uneven gene expression levels across different libraries (Fig 6C).

Interestingly, members of the viral families *Totiviridae* and *Narnaviridae*, known to naturally infect protozoa or fungi, showed co-occurrence with *K. yunnanensis* (Fig 2C). To explore this relationship, we analyzed the correlation between

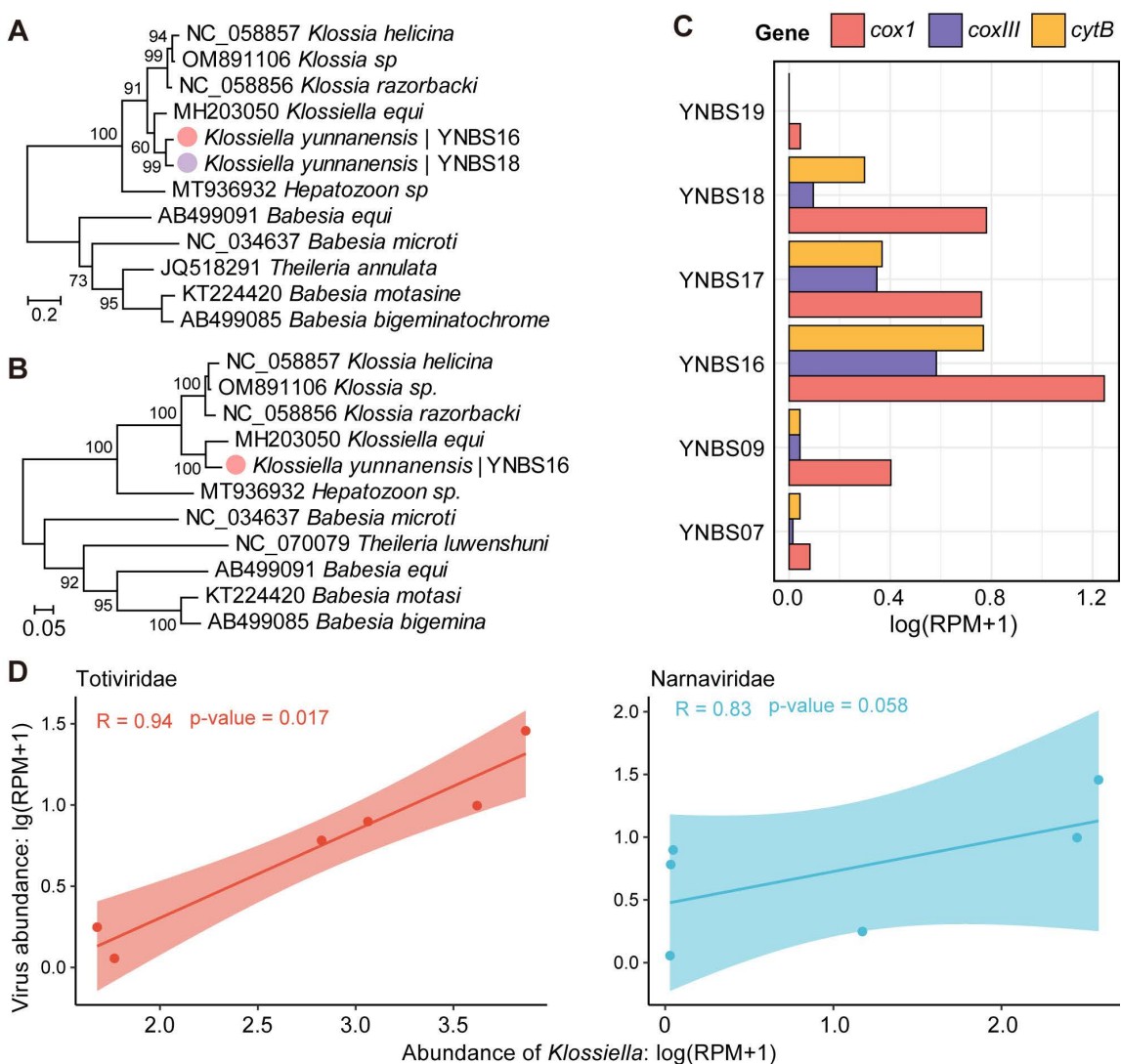

**Fig 6. Identification and characterization of a eukaryotic microbe.** (A, B) Phylogenetic trees of *Klossiella*, estimated using nucleotide sequences of the cox1 gene (a) and cytb gene (b). Colored dots indicate newly identified eukaryotic species, with colors corresponding to host genera. (C) Transcriptomic profiles of the *Klossiella* mitochondrion, represented as RPM, across positive pools. (D) Spearman's correlation analysis showing the relationship between the total relative abundance (RPM) of totiviruses, narnaviruses, and *Klossiella yunnanensis*.

the relative abundances (in RPM) of these viruses and *Klossiella* across the six positive pools. Strong positive correlations were observed, with Spearman's ρ ranging from 0.83 to 0.94 ($p < 0.05$) (Fig 6D), suggesting that the viruses in these families are likely hosted by *K. yunnanensis* rather than by bats.

## Discussion

There have been many studies examining the presence of viruses, bacteria, and eukaryotic microbes (i.e., fungi and protozoan parasites) in various bat tissues, including the brain, lung, liver, rectum, feces, urine, throat, and fecal swabs [9,13,31–34,37,38]. In contrast, the infectome composition of kidneys has received comparatively little attention. Our meta-transcriptomic sequencing of bat kidneys revealed a diverse array of microorganisms, shedding light on the broader bat infectome. Although viruses were the predominant microbial group identified, only 9 of the 22 detected viral species were categorized as mammalian viruses. Notably, the mammal-associated viruses identified in the kidneys differed from those identified in the rectal tissues of the same individual bats [13]. These findings align with previous research showing that viruses from different families exhibit marked variation in their organ-specific distribution in bats [14]. As a consequence, these results underscore the importance of adopting a multi-organ approach to comprehensively understand the microbial diversity harbored by bats, particularly for identifying host-microbe interactions. Furthermore, considering that kidney-associated pathogens, such as henipaviruses, may be shed through urine [39,40], future research should incorporate both kidney and urine sampling to comprehensively evaluate pathogen shedding and the associated transmission risk.

Of particular note, our study identified two novel henipaviruses that cluster within the bat-associated clade of this genus, including the lineage containing the Hendra and Nipah viruses. Nipah virus (NiV) are lethal pathogens that cause severe diseases in humans, including acute respiratory distress and encephalitis, with a mortality rate of 35–75% [35,41]. Similarly, Hendra virus (HeV) has caused multiple fatal outbreaks in humans and horses, including the death of veterinarians [35]. These viruses are naturally hosted by fruit bats (*Pteropus* species) and are typically transmitted to humans through bat urine or saliva, often via contamination of food sources [39,40]. HeV and NiV were first identified in Australia and Malaysia, respectively, and associated with *Pteropus* and other bat species [5,6]. In this study, we identified two related henipaviruses in *Rousettus leschenaultii* bats, marking the first detection of full-length henipavirus genomes in bats from China. This finding is particularly significant as Yunnan province is a recognized hotspot for bat diversity [31–34] and is located in southwestern China, representing the region of China geographically closest to Malaysia, where NiV first emerged (Fig 1A). Previously, antibodies to Nipah or Nipah-like viruses have been reported in bats from multiple regions in China, including Yunnan, Guangdong, Hainan and Hubei provinces, suggesting potential exposure to such viruses [42]. A recent large-scale study of bat RNA viral metagenomes detected genomic fragments of Nipah virus in bats from Southwest China [43], highlighting the potential circulation risk of Nipah-like viruses. However, these findings were not confirmed by RT-PCR, and the absence of full-length genomes further underscores the importance of our study, which provides genome-scale evidence for the diversity of henipaviruses in China and their zoonotic risk. Notably, more distantly related viruses have been discovered in rodents and shrews, including Mojiang virus [44] and Langya virus [45], with the latter confirmed to infect humans. These findings highlight the significance of the continued surveillance and genomic characterization of henipaviruses in bats, which are critical for understanding their potential spillover risk.

We also identified at least one bacterial species prevalent in bat kidneys. While the gut microbiota of bats has been extensively studied, less attention has been given to those of other organs, including the kidneys [16]. Previous research identified *Leptospira spp.* in bat kidney, supporting the hypothesis that bat kidneys may serve as a reservoir for zoonotic *Leptospira* [28,29,46,47], and we previously detected pathogenic *Leptospira* in bat kidneys using nested PCR in individual tissue samples [48]. However, no *Leptospira*-associated reads were detected in the meta-transcriptomic sequencing of this study, possibly due to sample pooling which might obscure the detection of low-abundance microbes. Instead, we identified *Flavobacterium* and *Phyllobacterium*, of which *Phyllobacterium calauticae* exhibited relatively high abundance and prevalence (Figs 2 and 5). *Phyllobacterium calauticae* is an aerobic, motile bacterium isolated from microaerophilic

freshwater sediments, adapted to efficiently utilize oxygen in low-oxygen environments [49]. This is not unprecedented, as *Listeria monocytogenes*, an environmentally ubiquitous bacterium, has previously been isolated from various wild animals, including bat kidneys [50,51]. Although these bacteria were relatively abundant, their biological significance in bats remains unclear. On one hand, their high abundance in tissue may suggest a pathogenic role. Alternatively, they may represent commensals or opportunistic colonizers, although their presence in the kidney is less probable.

Previous studies have shown that bats harbor a diverse range of protozoan parasites, some of which are capable of infecting humans [52]. However, there is only limited research on protozoan parasites present in bat kidneys. *Toxoplasma gondii*, a zoonotic protozoan parasite, has been detected in bats collected in Yunnan [53], and herein we identified a protozoan parasite, tentatively named *Klossiella yunnanensis*, in six (30%) of the libraries. Phylogenetic analysis suggests that *K. yunnanensis* is closely related to species known to infect horses, *K. equi*, which is generally considered non-pathogenic but can cause kidney alterations in cases of heavy infection [36]. The pathogenicity of this eukaryotic parasite to humans or even bats remains unclear. Moreover, the absence of a complete reference genome and the low sequence similarity to available references prevented the recovery of the full nuclear genome, preventing a more comprehensive genomic characterization of this parasite.

In addition, viruses from the families *Totiviridae* and *Narnaviridae* [54,55], which are known to infect a wide range of non-vertebrate host types including protozoan parasites, were detected in high abundance in bat kidneys. Spearman's correlation analysis of relative abundances indicated that these viruses were associated with *K. yunnanensis* rather than the bat hosts themselves (Fig 6D). This highlights the importance of conducting studies of the total infectome to better elucidate the interactions between viruses within an animal and their potential relationship with the primary host. Furthermore, we detected near-complete genomes of chuviruses in bat kidney tissues, which questions their usual assignment as viruses that do not infect vertebrates. While it is possible that these viruses infect eukaryotic microbes within the host, recent studies have increasingly identified chuviruses in mammals [14], suggesting that their host range may be broader than their previous designation as insect-specific viruses.

Our study has several limitations. Uneven sampling across locations and bat species—in which each species was sampled at only one or two sites—complicated our ability to assess virus distribution, compare viral compositions between species, and identify transmission networks. Although practical for broad surveys, pooling precludes resolving whether microbes originated from co-infections in single bats or from distinct individuals. Pooling may have also reduced sensitivity for detecting low-abundance microbes and potentially compromised the accuracy of microbial quantification. In addition, the lack of a complete genome assembly for the newly discovered eukaryotic parasite, coupled with the absence of a reference genome in existing databases, limited our ability to accurately quantify its abundance. The reliance on reference genomes from closely related species for abundance estimation may also introduce inaccuracies, particularly for divergent or poorly characterized taxa. These limitations collectively underscore the value of sequencing individual samples and integrating DNA-based approaches to obtain a more comprehensive view of the total infectome. Despite these limitations, our study offers the first comprehensive characterization of the bat kidney infectome, providing a foundation for more effective discovery and characterization of potential bat-borne pathogens.

## Materials and methods

### Ethics statement

This research, including the specimen collection and processing procedures, was reviewed and approved by the Ethics Committee of the Yunnan Institute of Endemic Disease Control and Prevention (File No. 20160002). All experiments were conducted with the approval of the Biosafety Committee of the same institute.

### Sample collection

Five sampling sites in Yunnan province were selected, denoted RL, ML, SB, LS, and JP (Fig 1A). From 2017 to 2021, bats were captured using mist nets (12 m × 2.5 m, mesh size 38 mm) deployed near caves and orchards during evening hours.

To minimize stress, nets were inspected every 15–20 minutes by trained personnel. Only bats displaying signs of weakened vital functions were selected for dissection. They were first humanely euthanized via intracardiac injection of sodium pentobarbitone and subsequently dissected for internal organ collection, following approved ethical protocols. Initial identification of bat species was performed by experienced field biologists based on morphological characteristics. Captured bats were then transported to the laboratory, euthanized by intracardiac delivery of sodium pentobarbitone, and dissected. Kidney tissues were collected and stored at −80 °C until further analysis. Preliminary species identification was confirmed by sequencing the cytochrome c oxidase I (*cox1*) gene for each specimen [56]. Mammalian species confirmation was achieved using *de novo* assembled *cox1* gene contigs. The final clean cox1 contigs were compared against the database within the BARCODE OF LIFE DATA SYSTEM (BOLDSYSTEMS) [57], and phylogenetic analyses were conducted using PHYML 3.0 [58] for species identification.

## Meta-transcriptomic sequencing

Individual tissues were initially organized into sample groups based on species identification and collection location. Specifically, 142 kidney tissues were grouped into 20 libraries, each comprising 2–8 individuals (S1 Table). Total RNA was extracted and purified from each pool using the RNeasy Plus Universal Mini Kit (Qiagen, Germany). RNA libraries were constructed using the Zymo-Seq RiboFree Total RNA Library Kit (No. R3003) (Zymo Research, USA), following the manufacturer's instructions. These libraries were sequenced using paired-end 150 bp reads on the Illumina NovaSeq 6000 sequencing platform.

## Characterization of total infectomes

Adapter sequences were removed from the sequencing reads, and initial quality control was performed using the pipeline implemented in bbduk.sh (https://sourceforge.net/projects/bbmap/). Duplicate reads were filtered out using cd-hit-dup with default settings [59]. rRNA reads were removed by mapping the processed reads against the SILVA rRNA database (Release 138.1) using Bowtie2 (version 2.3.5.1) in '--local' mode [60]. The remaining high-quality, non-rRNA reads were either (i) directly compared against the non-redundant protein (nr) database using DIAMOND BLASTx [61], or (ii) assembled into contigs using MEGAHIT (version 1.2.8) [62] before comparison against the National Center for Biotechnology Information (NCBI) non-redundant protein (nr) database. An e-value threshold of $1 \times 10^{-5}$ was set to maintain high sensitivity and minimize false positives.

For virus identification, contigs identified from the kingdom 'Viruses' were extracted, and those shorter than 600 bp were excluded to ensure the quality of virus genomes. The remaining overlapping contigs were merged into extended viral sequences using the SeqMan program implemented in the Lasergene software package version 7.1 (DNAstar, USA) [63]. To assign species-level classifications, all viral contigs were clustered using CD-HIT (v4.8.1) [59], applying identity thresholds based on ICTV species demarcation criteria for the corresponding viral genera [55]. Representative contigs from each cluster were then compared to known viral species to determine whether they matched previously recognized or potentially novel viruses. For genera lacking explicit species demarcation criteria, a 90% amino acid identity threshold for the RdRP or replicase protein was applied (S2 Table). The abundance of these viral contigs was estimated by mapping reads back to the assembled genomes using Bowtie2 version 2.5.2 with '--end-to-end' and '--very-fast' settings. Reads mapped to all contigs assigned to the same viral species were aggregated to calculate the final abundance of that species in each library. Alignments were sorted and indexed with SAMtools version 1.18 and visualized with Geneious Prime version 2020.2.4 [64,65].

For bacteria and eukaryotic microbes, we initially utilized MetaPhlAn version 4 to identify potential microbial taxonomy [66]. We then performed a *de novo* assembly of the reads using MEGAHIT (version 1.2.8) as described above [62]. Assembled contigs were compared against conserved bacterial marker genes (e.g., *rpoB*, *groEL*, *recA*, and *gyrB*) and

eukaryotic microbial genes (e.g., *EF1-alpha*) using DIAMOND BLASTx [61]. Complete reference genome sequences of the corresponding bacterial and protozoan genera were subsequently downloaded from GenBank and used as templates for read mapping and gene abundance estimation with Bowtie2 (version 2.5.2) [60]. Highly conserved regions, such as rRNA genes, were excluded from the reference genome sequences before conducting mapping analyses. From the aligned reads, we generated consensus sequences for well-covered protein-coding regions, with a focus on phylogenetically informative loci such as *rpoB* and *groEL*, which provide high species-level resolution in bacterial systematics. Finally, these consensus sequences were subsequently subjected to BLASTn comparisons against the NCBI nucleotide (nt) database to determine microbial taxonomy at the species level.

## Evolutionary analyses

To determine the evolutionary relationships of the newly identified microbes, reference nucleotide/amino acid sequences for microbial taxa in question were downloaded from the NCBI GenBank Database. In all cases, sequences were then aligned using MAFFT [67], with the 5' and 3' unaligned regions (when present) removed manually and ambiguously aligned sequences excluded using TrimAl version 1.5.0 [68]. Phylogenetic trees on these data were then estimated using the maximum likelihood method implemented in PHYML 3.0, employing the GTR model of nucleotide substitution and SPR branch swapping [58]. Node support was estimated using an approximate likelihood ratio test using Shimodaira–Hasegawa-like procedures.

## Characterization of henipaviruses

To assess the prevalence of novel henipaviruses in bats and in different organs, real-time quantitative reverse transcription PCR (qRT-PCR) and nested RT-PCR were performed on all individual kidney samples. Specific primers were designed using the virus genome sequences obtained from libraries YNBS03 (Yunnan bat henipavirus 1) and YNBS02 and YNBS04 (Yunnan bat henipavirus 2). To investigate viral distributions across various bat organs, PCR detection and individual meta-transcriptomics assays were performed on the brain, heart, liver, kidney, and gut sample of the positive bats (WD1733 and WD1745). However, the library construction for the brain sample from WDBN1733 failed.

As the full-length sequence of Yunnan bat henipaviruses 1 was not initially obtained, PCR assays and Sanger sequencing were employed to complete it. The final genome consensus sequences were confirmed by mapping the reads against draft genome sequences, and viral abundance was estimated based on the number of reads mapped to genome [60]. For each complete genomes, potential open reading frames (ORFs) and coding arrangements were predicted using ORFfinder (https://www.ncbi.nlm.nih.gov/orffinder/) and annotated by blastp program (https://blast.ncbi.nlm.nih.gov/Blast.cgi). Phylogenetic trees for each gene were estimated following the standard protocol described above.

## Supporting information

**S1 Fig. Maximum likelihood phylogenetic tree estimated using amino acid sequences of the DNA polymerase within the genus *Orthohepadnavirus* (*Hepadnaviridae*).** The newly identified virus in this study is marked with a solid red circle. Bat-derived viruses and their corresponding clades—determined according to divergence levels used by ICTV as species demarcation criteria—are labeled on the right.
(TIF)

**S1 Table. Information of sample group and RNA library in this study.**
(XLSX)

**S2 Table. Viruses in bat kidneys identified in this study.**
(XLSX)

**S3 Table. Summary of contigs with BLASTx hits to conserved bacterial marker genes.**
(XLSX)

## Acknowledgments

We wish to thank the local Centers for Disease Control and Prevention in five trapping sites for their assistance in specimen collection.

## Author contributions

**Conceptualization:** Guopeng Kuang, Tian Yang, Weihong Yang, Hong Pan, Yao-qing Chen, Edward C. Holmes, Mang Shi, Yun Feng.

**Data curation:** Guopeng Kuang, Tian Yang, Weihong Yang, Hong Pan, Mang Shi, Yun Feng.

**Formal analysis:** Guopeng Kuang, Tian Yang, Weihong Yang, Hong Pan, Yun Feng.

**Funding acquisition:** Weihong Yang, Mang Shi, Yun Feng.

**Investigation:** Guopeng Kuang, Tian Yang, Weihong Yang, Jing Wang, Hong Pan, Wei-chen Wu, Juan Wang, Lifeng Yang, Xi Han, Yao-qing Chen, John-Sebastian Eden, Mang Shi, Yun Feng.

**Methodology:** Guopeng Kuang, Tian Yang, Weihong Yang, Jing Wang, Hong Pan, Yuanfei Pan, Qin-yu Gou, Wei-chen Wu, Juan Wang, Lifeng Yang, Xi Han, Yao-qing Chen, Mang Shi, Yun Feng.

**Project administration:** Guopeng Kuang, Tian Yang, Weihong Yang, Jing Wang, Mang Shi, Yun Feng.

**Resources:** Guopeng Kuang, Tian Yang, Weihong Yang, Jing Wang, Wei-chen Wu, Juan Wang, Lifeng Yang, Xi Han, Yun Feng.

**Software:** Guopeng Kuang, Tian Yang, Jing Wang, Yuanfei Pan, Qin-yu Gou, Wei-chen Wu, Mang Shi.

**Supervision:** Yuanfei Pan, Qin-yu Gou, Wei-chen Wu, John-Sebastian Eden, Mang Shi, Yun Feng.

**Validation:** Guopeng Kuang, Tian Yang, Yuanfei Pan, Qin-yu Gou, Mang Shi, Yun Feng.

**Visualization:** Guopeng Kuang, Tian Yang, Yuanfei Pan, Qin-yu Gou, Mang Shi.

**Writing – original draft:** Guopeng Kuang, Tian Yang, Edward C. Holmes, Mang Shi, Yun Feng.

**Writing – review & editing:** Guopeng Kuang, Tian Yang, Weihong Yang, Jing Wang, Hong Pan, Yuanfei Pan, Qin-yu Gou, Wei-chen Wu, Juan Wang, Lifeng Yang, Xi Han, Yao-qing Chen, John-Sebastian Eden, Edward C. Holmes, Mang Shi, Yun Feng.

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
