## [Decision Letter · Decision Letter 0]

19 Apr 2025

PPATHOGENS-D-25-00320

Infectome analysis of bat kidneys from Yunnan province, China, reveals close relatives of Hendra-Nipah viruses and prevalent bacterial and eukaryotic pathogens

PLOS Pathogens

Dear Dr. Feng,

Thank you for submitting your manuscript to PLOS Pathogens. After careful consideration, we feel that it has merit but does not fully meet PLOS Pathogens's publication criteria as it currently stands. Therefore, we invite you to submit a revised version of the manuscript that addresses the points raised during the review process.

Please submit your revised manuscript within 30 days Jun 18 2025 11:59PM. If you will need more time than this to complete your revisions, please reply to this message or contact the journal office at plospathogens@plos.org. Please include the following items when submitting your revised manuscript:

We look forward to receiving your revised manuscript.

Kind regards,

Darren J. Obbard

Guest Editor

PLOS Pathogens

Ronald Swanstrom

Section Editor

PLOS Pathogens

Sumita Bhaduri-McIntosh

Editor-in-Chief

PLOS Pathogens

orcid.org/0000-0003-2946-9497

Michael Malim

Editor-in-Chief

PLOS Pathogens

orcid.org/0000-0002-7699-2064

**Additional Editor Comments:**

Thank you for your submission. As you will see, all three reviewers agree on the value in this work, and feel that - overall - it is very well done. The manuscript is interesting, and relatively easy to read, and the novel henipavirus give the paper quite a broad level of interest.

In any revision, I ask that you take on board all of the reviewers' comments. I suspect the vast majority of comments should be relatively easy to address through textual changes (e.g. more stringent use of the word 'pathogen') or through a change in framing. While I would welcome more detailed analysis of non-viral microbiota and any further validation that might be possible, given the scale of the manuscript as it stands I recognise that further work along these lines might be beyond the scope of what is feasible.

Nevertheless, in addition to the scientific / analytical suggestions, I particularly want to draw your attention to the absolute need for

(1) A clear statement on the methodology and nets used for trapping (mist nets, sticky nets?) and how the approach used conforms to the relevant ethical standards,

and (2) Full availability of all raw and processed sequences under PRJNA1184956 - which is not yet publicly available.

**Journal Requirements:**

1) Please upload all main figures as separate Figure files in .tif or .eps format. For more information about how to convert and format your figure files please see our guidelines: 

2) Please ensure that the funders and grant numbers match between the Financial Disclosure field and the Funding Information tab in your submission form. Note that the funders must be provided in the same order in both places as well.

**Reviewers' Comments:**

Reviewer's Responses to Questions

**Part I - Summary**

Reviewer #1: The manuscript by Kuang et al. investigates the total infectome present in bat kidneys collected from Yunnan, identifying close relatives of highly significant pathogens, including henipaviruses—BSL-4 level viruses—alongside abundant bacteria and parasites. While the study is limited in scale in terms of the number of animals and geographic coverage, its findings are highly valuable and of great interest to the virology community. Notably, the discovery of bat-associated henipaviruses is both novel and significant. Previous studies were either based solely on serological evidence or lacked reliability due to the absence of PCR confirmation. Another key strength of this work is its novel approach in characterizing the total infectome, rather than focusing solely on virome or specific viral families like coronaviruses, providing a more comprehensive perspective.

Reviewer #2: This article presents interesting data on the infectome of the kidneys of 10 different bat species in five regions of Yunnan Province in China (Fig1). New viruses, including some with zoonotic potential (henipaviruses), as well as a novel protozoan and 2 bacteria, were identified using a meta-transcriptomic approach (Fig2 and 3). The two henipaviruses were subsequently confirmed by qRT-PCR and nested RT-PCR testing of different tissues to determine virus distribution within 2 bats, one testing positive in the lung and kidney and the other testing positive in all tissues (heart, liver, kidney, gut) except the brain. The newly identified henipavirus 1 was present in 1 of the 20 libraries, and the henipavirus 2 was present in 2 of the 20 libraries, both corresponding to Rousettus leschenaultii and collected near areas with fruit orchards. Individual bat testing identified 4 bats carrying these henipaviruses, and within these bats, several tissues tested positive for the virus with the kidney showing the highest sequence yield (supplemental material), suggesting this is the target tissue tropism for these viruses. Characterization of the ORFs, identified the sequences for the 6 proteins found in henipaviruses, and phylogenetic analysis grouped these new viruses with other bat henipaviruses, including the zoonotic Hendra and Nipah. Observation of variable clustering depending on which of the 6 genes was being evaluated suggests recombination events (Fig4). The 2 novel bacterial species described include a Flavobacterium (94% nucleic acid identity in the ropB gene) and a Phyllobacterium (97% nucleic acid identity in the groEL gene) (Fig 5). The rationale for the selection of these genes was unclear, and the relevance of these findings was only briefly mentioned. It was noted that bacteria such as Leptospira were expected to be found in the kidneys of these bats, but weren’t. The protozoan from the Klossiellidae (Fig6) was present in 6 pools, and analysis of the COX1 and the cytB genes resulted in high similarity to Klosiella equi, an organism not usually considered pathogenic. The bat species identification was done by morphological characteristics and confirmed by genetic testing of COX1. Bats were pooled into 20 groups with variable numbers (2-8 individuals) for sequencing library construction. The authors acknowledge the drawback of sample pooling (lines 264 and 290) but also demonstrate the value of this approach. Two of the 20 libraries rendered no microbial presence. Library preparation, sequence curation, and phylogenetic analysis protocols described in the materials and methods seemed appropriate. The results of this study provide valuable information: The results indicated a preponderance of RNA viruses (20 species within 12 families), one DNA virus, one reverse-transcribing virus, one protozoan, and two abundant bacterial species (one of them present in 65% of the libraries). Nine of the virus species were phylogenetically related to known mammalian viruses. The methodology is consistent with other studies, the manuscript is well-organized and written, and the figures are clear.

Reviewer #3: Kuang and colleagues have investigated and reported the infectome of bat kidneys sampled in Yunnan Province, China. This manuscript is both timely and relevant, as it provides further evidence of novel microorganisms that infect wild animals, demonstrating specific tissue tropism and tissue specific viromes. The findings are particularly significant due to the identification of a divergent paramyxovirus that clusters within the same clade as lethal human pathogens such as Nipah and Hendra viruses. However, while the results are robust, the authors tend to overstate the pathogenic potential of the microorganisms identified. Throughout the manuscript, the microorganisms are frequently referred to as pathogens, despite the absence of experimental or indirect evidence—such as observed disease signs in animals—to support this classification. Follow additional comments below.

**Part II – Major Issues: Key Experiments Required for Acceptance**

Reviewer #1: 1. The authors should reference the recent publication in Nature Microbiology by He et al., which also reports the discovery of Nipavirus-like viruses in China. Although that study only identified viral fragments, and the authors of this manuscript failed to obtain longer genomic sequences using PCR, some discussion comparing the findings would be valuable.

2. The authors have done a thorough job in characterizing henipaviruses, making full use of the available data. However, the characterization of bacteria is relatively limited. While I understand that these may be of lesser priority, it would be valuable to perform single-library RNA and DNA sequencing on the samples with highest abundance of bacteria to obtain draft genomes. Any novel pathogens would merit revealing full genomes.

3. Similarly, the transcriptomic characterization of parasites is currently limited to mitochondrial genes—what about nuclear genes? Expanding this analysis would provide a more comprehensive understanding of the parasite component in the infectome.

Reviewer #2: Some clarification on the nets used for trapping is necessary. Can the type of net and source be provided?

Reviewer #3: Page 4 line 57 - The authors stated “broader pathogen surveillance beyond the gastrointestinal tract” A pathogen per definition is a microorganisms that induce disease. The authors have no data to infer pathogenicity even for the new henipaviruses. I recommend the authors to moderate their claims throughout the manuscript.

Data availability:

PRJNA1184956 - I could not find the raw dataset at NCBI SRA. Neither the consensus genomic data with the accession numbers provided.

Page 16 line 352-360 - Does the authors consider that mapping reads to reference bacterial and fungi genomes can severely bias the results? For the taxa that are closely related to reference genomes it should be ok once reads will map against with high confidence, but for more diverging taxa reads will not even map since Bowtie 2 has a limited number of snps allowed to map a read against a reference sequence. As pointed out by the authors, there is no study investigating bat kidneys using metagenomics/metatranscriptomics. Therefore, most of the bacterial and fungi species likely infecting these bats may be substantially divergent from the reference genomes available in the databases. I recommend to the authors to take a more holistic view of bacteria and fungi performing Diamond search on the assembled contigs from Megahit and extracting the taxonomic identification from it.

**Part III – Minor Issues: Editorial and Data Presentation Modifications**

Reviewer #1: 4. Terms like “infectome” should be defined in the abstract as well.

5. The geographic locations and ecological factors (in any) of the newly found viruses should be discussed.

Reviewer #2: The references are adequate, but there are at least 3 references that are repeated (ref 21 is a repeat of 19, 49 is a repeat of 29, and 50 is a repeat of 30).

Reviewer #3: Abstract

How the authors can ascertain if the protozoan and bacteria found are relevant for infection/pathogenicity, maybe they are just highly prevalent non deleterious symbionts

Page 5 line 79-80 - The authors stated “While much of this research has focused on the bat gut virome, viruses present in other body sites—such as the kidneys—also pose transmission risks.” Those viruses present in the kidney are not supposed to be excreted through urine? That is the way Nipah is excreted and gets in contact with humans through a natural beverage contaminated with urine carrying infectious Nipah viral particles, for instance. One thing that may impact the chance of detecting the virus in the urine is that the viral particle is perhaps labile and degrades quickly and viral loads may not be that high compared with kidney tissue. An interesting strategy to sample wild bats and test this hypothesis which also can provide critical data to evaluate if the virus is being excreted is to sample bat urine and only after euthanasiate the bat to collect the kidney. But very good sampling procedures are required to avoid cross contamination of samples of the same bat in the field.

Figure 1 A - I think it would be worthwhile for the readers to see the Yunnan province contextualized within China and bordering countries. Besides, if the authors can also depict it including the geography and biome of the region it is interesting for the readers to understand bats species and populations residing in these environments and why they are more likely related to bat species and population of neighbouring countries and not to other Chinese provinces. Providing the context of neighbouring countries is particularly important because Nipah outbreaks have been reported in neighbouring countries.

Page 7 line 112 - “Based on mitochondrial sequences and sampling locations, the samples were pooled into 20 groups for” - Does it mean that samples from the same species and caves/sites were pooled together? What about samples of the same population from different time points or there were no resampling of the same population in different time points?

Figure 2 B - This is a very difficult graph to interpret. I suggest the authors look for an alternative way to show these results. In fact, these results are already shown in much more detail in section C. It is not the same metric, but the presence of viruses per taxonomic level distributed per host species is already there in much more detail.

Figure 2 B- How the authors determined that, for instance, Yannan bat henipavirus 2 what was present in different pools of H. armiger? Did these henipaviruses show high sequence identity? I have not seen a methodological description of how the authors approached the detection of the same virus in multiple samples in the material and methods. The only section about this regards the use of ICTV parameters, but it would be important to provide more detail stating for instance, that “we considered the same virus present in multiple samples when ???? clustered together in the phylogenetic tree and showed aa ou nucl identity of ??? in the entire genome??”

It is interesting that the authors found Chuvirus and other viruses in the kidney tissue of several different bat species and they did not comment on the possibility of these viruses to infect mammals, even though the current knowledge suggests that these viruses have other animals taxa (invertebrates etc) as main hosts. Finding a full genome of a virus within the kidney suggests that the virus may be infecting that tissue. On the other hand, it is hard to associate the presence of such viruses as of dietary sources as pointed out in page 8 line 154-155. Except for the Toti and Narna comments by the authors on Page 11 line 217.223.

Page 8 line 142-145 - amino acid ou nucleotide identity? L protein I assume amino acid. This is a substantial difference.

Page 10 line 193 - “likely reflects the action of recombination.“ or due to the high divergence it may represent the accumulation of several mutations leading to the phylogenetic signal saturation leaving few informative sites that allows more consistent phylogenetic grouping inferences. Please, add alternative explanations even more considering that the data is not suitable to evaluate recombination (divergent viruses).

Page 10 line 197 - “bacterial pathogens” How the authors know that these are pathogenic bacteria, this same question also applies to viruses and fungi found? Were there any signs of disease in the sampled bats? Although there is some indirect evidence that some may be pathogens (belonging to a family/taxa that include several known pathogens and are phylogenetically related to known pathogens) there is no other supporting evidence to suggest pathogenicity in this manuscript.

Page 12 line 239 - “our study identified close relatives”. Close relatives is a bit strong and arbitrary adjective based on the amino acid divergence and instability of the phylogenetic positioning depending on the protein used for phylogenetic reconstruction. I suggest the authors moderate it. The results are important and interesting, but those are not close relatives of Hendra and Nipah.

Page 12 line 246 - italicize Pteropus

Page 13 line 278-9 - “The pathogenicity of these eukaryotic parasites to humans or even bats remains unclear.” Here the authors emphasized that more studies are necessary to ascertain if the identified microbes are pathogenic. But I think the authors must review more throughout the wording and tone throughout the manuscript as pointed out multiple times above including in the title.

PLOS authors have the option to publish the peer review history of their article (what does this mean? ). If published, this will include your full peer review and any attached files.

**Do you want your identity to be public for this peer review?** For information about this choice, including consent withdrawal, please see our Privacy Policy .

Reviewer #1: No

Reviewer #2: No

Reviewer #3: No

**Figure resubmission:**
---

## [Editor Report · Decision Letter 1]

26 May 2025

Dear Dr. Feng,

We are pleased to inform you that your manuscript 'Infectome analysis of bat kidneys from Yunnan province, China, reveals novel henipaviruses related to Hendra and Nipah viruses and prevalent bacterial and eukaryotic microbes' has been provisionally accepted for publication in PLOS Pathogens.

Best regards,

Darren J. Obbard

Guest Editor

PLOS Pathogens

Ronald Swanstrom

Section Editor

PLOS Pathogens

Sumita Bhaduri-McIntosh

Editor-in-Chief

PLOS Pathogens

orcid.org/0000-0003-2946-9497

Michael Malim

Editor-in-Chief

PLOS Pathogens

orcid.org/0000-0002-7699-2064

Thank you for your positive and wholehearted engagement with my and the reviewers' comments and questions. I do not see any further scientific or presentation issues in need of response.
---

## [Editor Report · Acceptance letter]

Dear Dr. Feng,

We are delighted to inform you that your manuscript, "Infectome analysis of bat kidneys from Yunnan province, China, reveals novel henipaviruses related to Hendra and Nipah viruses and prevalent bacterial and eukaryotic microbes," has been formally accepted for publication in PLOS Pathogens.

Best regards,

Sumita Bhaduri-McIntosh

Editor-in-Chief

PLOS Pathogens

orcid.org/0000-0003-2946-9497

Michael Malim

Editor-in-Chief

PLOS Pathogens

orcid.org/0000-0002-7699-2064